# Cross-sectional analysis of UK research studies in 2015: results from a scoping project with the UK Health Research Authority

Tim Clark,[1] Richard H Wicentowski,[2] Matthew R Sydes[3]

¹Faculty of Medicine, Institut für Medizinische Informationsverarbeitung, Biometrie und Epidemiologie (IBE), Ludwig-Maximilians University, Munich, Germany
²Computer Science Department, Swarthmore College, Swarthmore, Pennsylvania, USA
³MRC Clinical Trials Unit at UCL, Institute of Clinical Trials and Methodology, University College London, London, UK

**Correspondence to**
Mr Matthew R Sydes;
m.sydes@ucl.ac.uk

## ABSTRACT

**Objectives** To determine whether data on research studies held by the UK Health Research Authority (HRA) could be summarised automatically with minimal manual intervention. There are numerous initiatives to reduce research waste by improving the design, conduct, analysis and reporting of clinical studies. However, quantitative data on the characteristics of clinical studies and the impact of the various initiatives are limited.

**Design** Feasibility study, using 1 year of data.

**Setting** We worked with the HRA on a pilot study using research applications submitted for UK-wide ethical review. We extracted into a single dataset, information held in anonymised XML files by the Integrated Research Application System (IRAS) and the HRA Assessment Review Portal (HARP). Research applications from 2014 to 2016 were provided. We used standard text extraction methods to assess information held in free-text fields. We use simple, descriptive methods to summarise the research activities that we extracted.

**Participants** Not applicable—records-based study

**Interventions** Not applicable.

**Primary and secondary outcome measures** Feasibility of extraction and processing.

**Results** We successfully imported 1775 non-duplicate research applications from the XML files into a single database. Of these, 963 were randomised controlled trials and 812 were other studies. Most studies received a favourable opinion. There was limited patient and public involvement in the studies. Most, but not all, studies were planned for publication of results. Novel study designs (eg, adaptive and Bayesian designs) were infrequently reported.

**Conclusions** We have demonstrated that the data submitted from IRAS to the HRA and its HARP system are accessible and can be queried for information. We strongly encourage the development of fully resourced collaborative projects to further this work. This would aid understanding of how study characteristics change over time and across therapeutic areas, as well as the progress of initiatives to improve the quality and relevance of research studies.

## INTRODUCTION

The need to improve the quality of clinical research is increasingly understood.[1] With

**Strengths and limitations of this study**

► First study to draw information from Health Research Authority (HRA's) Assessment Review Portal HARP system into one searchable dataset.
► Anonymised data from the HRA's HARP system can be interrogated with minimal manual intervention.
► Feasibility study, so limited to data from 2015 only and excluding phase I healthy volunteer studies.
► Aligning the questions from the XML document was not straightforward.
► The search terms chosen for the free-text fields were not exhaustive and because it was unfeasible in a pilot to review all applications, the sensitivity and specificity of the text-mining methodology could not be calculated.

a particular focus on clinical studies, the 'gold standard' for evidence-based medicine, there has been a noticeable push towards: improving study protocols[1]; developing and implementing newer methodologies, such as adaptive designs[2]; involving patients in the design, conduct and management of studies[3]; and ensuring that study results are quickly and accurately reported.[4]

In order to properly evaluate the current state of clinical research and changes over time, it is clearly necessary to have unfettered access to the research protocols and study results.[5 6] There are a number of limitations in accessing such information. Published evaluations of the state of clinical research are mainly based on the clinical study publication due to the difficulty in obtaining access to unpublished research protocols.[7] Research registers are incomplete for many jurisdictions and/or are, like study publications, limited in detail compared with the research protocol. For example, clinicaltrials.gov[8] has good coverage of clinical studies in North America, where registration of clinical trials is mandatory. The WHO International

Clinical Trials Registry Portal[9] provides access to 15 different regional or national registries, but both have markedly less detail than in the study's research protocol. Published evaluations of the literature are likely affected by publication bias and there are often discrepancies between research publications and their underpinning protocols.[10]

The ability to analyse a large database of application forms, each of which contain very detailed information taken directly from the research protocol, would enable researchers to perform a detailed examination of the characteristics of clinical studies, particularly those necessary for generating reliable evidence. This would aid understanding of how study characteristics change over time and across therapeutic areas, as well as the progress of initiatives to improve the quality and relevance of clinical studies.[11–15] Therefore, we approached the UK Health Research Authority (HRA), which oversees national ethical review of health research conducted within the National Health Service (NHS). With the support of the HRA and having signed the appropriate confidentiality declarations, we developed a pilot project to determine the feasibility of interrogating with minimal manual intervention the data contained in HARP (HRA Assessment Review Portal). HARP is a web-based management information system used by the HRA for all research ethics applications that require NHS REC (National Health Service Research Ethics Committee) review and/or HRA approval. We negotiated access to data on clinical studies submitted in 2015. The extract had personal data and organisational identifiers removed by HRA and did not include phase I healthy volunteer studies.

Our pilot focused on automatically summarising the characteristics of clinical studies, including therapeutic area, blinding, randomisation, use of Independent Data Monitoring Committee (IDMC), patient and public involvement (PPI) in the research, dissemination of the study results and use of new methodologies such as adaptive designs.

## METHODS
The HRA collects applications for REC 'approval' (a favourable opinion) through the Integrated Research Application System (IRAS)[16] and stores them in HARP. IRAS is a web-based system used to capture the information required by review bodies in the UK, including the Medicines and Healthcare Products Regulatory Agency and HRA. When the forms are completed, IRAS saves the results as an XML document. XML is a commonly used file format that stores structured data in plain text. The XML document can be used to generate PDF copies of the required forms and to repopulate the online web tool so users can later edit their applications.

Once the appropriate permissions were agreed, the HRA provided us with anonymised applications for clinical studies submitted for review from 2014 to 2016, excluding phase I healthy volunteer studies. For this

pilot, we wished to focus on the extraction of clinical study characteristics of interest from the PDF versions of the HRA forms underlying the IRAS application system. We were provided with an incomplete mapping between the questions on the PDF and the data stored in the XML document, such that many questions relevant to our study were not immediately labelled; nor was it apparent which questions were effectively switched on and off for applicants in completing the system's initial filter questions for their application. Therefore, we reverse-engineered the structure of the data in IRAS to determine which XML tags were responsible for creating the content we wanted to extract from the XML files to study.

The IRAS tool has been updated over time; not all of applications were created using the same version. Although the majority (91%) of the XML documents were created with one version (the latest version of the tool in the dataset), eight different versions were present in the dataset we received. We were not provided with the documentation detailing the ways in which the versions differed; we had access only to the latest version. Our inspection of the data indicated that the differences relevant to this pilot study were minor and we corrected for these when they were detected; however, there may be some undetected differences where the distinction between versions was important for our study.

For our pilot project, we chose to collect information on: the number of randomised controlled trials (RCT); the trial phase; the therapeutic area; the number of clinical trials of investigational medicinal products (CTIMPs — trials that require registration with regulatory body); the research method used (eg, RCT, feasibility/pilot study, blinding); the use of blinding; the use of systematic or formal literature reviews in planning the study; the use of adaptive or Bayesian designs; the use of a data monitoring committee; the plans for dissemination of findings; and the role of PPI in aspects of the research. Success would be measured in terms of our ability to access the information and generate descriptive summaries with minimal need for manual data reviews.

The data for our target areas were either held in multiple choice questions that we could identify and summarise or in free-text fields from which information needed to be determined and extracted. For those questions, we preprocessed the text using Spacy (https://spacy.io/), a natural language processing (NLP) package for Python. This involved identifying sentence boundaries and performing morphological analysis to convert words into their canonical dictionary form. We then used regular expressions to match phrases we were interested in, ignoring differences relating to spacing, hyphenation and capitalisation; Roman numerals were standardised to Latin numbers. For example, when looking for evidence that a systematic review was either performed or an existing review used during the study planning process, we searched for phrases such as 'systematic review', 'reviewed systematically', 'literature review', 'evidence review' and 'evidence-based review' in the answers provided to

**Table 1** Overview of research activities submitted for REC approval

| Characteristic | RCT | | Other | | Total | |
|---|---|---|---|---|---|---|
| | N | % | N | % | N | % |
| N | 963 | | 812 | | 1775 | |
| **Therapeutic area*** | | | | | | |
| Cancer | 168 | 17 | 192 | 24 | 360 | 20 |
| Cardiovascular | 94 | 10 | 94 | 12 | 188 | 11 |
| Musculoskeletal | 97 | 10 | 30 | 4 | 162 | 9 |
| Respiratory | 97 | 10 | 30 | 4 | 159 | 9 |
| Paediatrics | 56 | 6 | 30 | 4 | 148 | 8 |
| Neurological | 73 | 8 | 30 | 4 | 145 | 8 |
| Mental health | 66 | 7 | 30 | 4 | 117 | 7 |
| Inflammatory and immune system | 72 | 7 | 30 | 4 | 113 | 6 |
| Oral and gastrointestinal | 64 | 7 | 30 | 4 | 113 | 6 |
| Blood | 44 | 5 | 67 | 8 | 111 | 6 |
| Diabetes | 58 | 6 | 38 | 5 | 96 | 5 |
| Infection | 55 | 6 | 30 | 4 | 92 | 5 |
| Renal and urogenital | 39 | 4 | 30 | 4 | 82 | 5 |
| Generic health relevance | 46 | 5 | 30 | 4 | 76 | 4 |
| Metabolic and endocrine | 32 | 3 | 30 | 4 | 70 | 4 |
| Dementias and neurodegenerative | 27 | 3 | 36 | 4 | 63 | 4 |
| Skin | 36 | 4 | 30 | 4 | 57 | 3 |
| Reproductive health and childbirth | 30 | 3 | 30 | 4 | 56 | 3 |
| Stroke | 27 | 3 | 30 | 4 | 53 | 3 |
| Eye | 24 | 2 | 28 | 3 | 52 | 3 |
| Injuries and accidents | 22 | 2 | 30 | 4 | 36 | 2 |
| Congenital disorders | 10 | 1 | 20 | 2 | 30 | 2 |
| Ear | 6 | 1 | 7 | 1 | 13 | 1 |
| **Count of therapeutic areas claimed** | | | | | | |
| None | 52 | 5 | 49 | 6 | 101 | 6 |
| 1 | 660 | 69 | 495 | 61 | 1155 | 65 |
| 2 | 201 | 21 | 200 | 25 | 401 | 23 |
| 3 | 29 | 3 | 49 | 6 | 78 | 4 |
| ≥4 | 21 | 2 | 19 | 2 | 40 | 2 |
| **Research methods used*** | | | | | | |
| RCT | 963 | 100 | 2 | 0 | 965 | 54 |
| Feasibility/pilot study | 177 | 18 | 300 | 37 | 477 | 27 |
| Questionnaire | 98 | 10 | 139 | 17 | 237 | 13 |
| Cohort observation | 23 | 2 | 161 | 20 | 184 | 10 |
| Qualitative research | 90 | 9 | 76 | 9 | 166 | 9 |
| Controlled trial, no randomisation | 3 | 0 | 157 | 19 | 160 | 9 |
| Laboratory study | 30 | 3 | 47 | 6 | 77 | 4 |
| Case series/case note review | 7 | 1 | 45 | 6 | 52 | 3 |
| Cross-sectional study | 4 | 0 | 31 | 4 | 35 | 2 |
| Case–control study | 5 | 1 | 25 | 3 | 30 | 2 |
| Database analysis | 8 | 1 | 17 | 2 | 25 | 1 |
| Epidemiology | 7 | 1 | 5 | 1 | 12 | 1 |

**Table 1** Continued

| Characteristic | RCT | | Other | | Total | |
|---|---|---|---|---|---|---|
| | N | % | N | % | N | % |
| Meta-analysis | 0 | 0 | 0 | 0 | 0 | 0 |
| Other | 68 | 7 | 269 | 33 | 337 | 19 |
| **Count of research methods used** | | | | | | |
| 1 | 641 | 67% | 522 | 64 | 1163 | 66 |
| 2 | 192 | 20 | 160 | 20 | 352 | 20 |
| 3 | 73 | 8 | 97 | 12 | 170 | 10 |
| 4 | 48 | 5 | 25 | 3 | 73 | 4 |
| 5 | 7 | 1 | 7 | 1 | 14 | 1 |
| 6 | 2 | 0 | 1 | 0 | 3 | 0 |
| **CTIMP** | | | | | | |
| Unlicensed | 315 | 33 | 189 | 23 | 504 | 28 |
| New use | 146 | 15 | 70 | 9 | 216 | 12 |
| Within SmPC | 66 | 7 | 26 | 3 | 92 | 5 |
| Other | 10 | 1 | 5 | 1 | 15 | 1 |
| *Any of these* | *515* | *53* | *284* | *35* | | |
| **Involves ionising radiation** | | | | | | |
| Yes | 666 | 69 | 582 | 72 | 1248 | 71 |
| No | 297 | 31 | 222 | 28 | 519 | 29 |
| *Missing* | 0 | NA | 8 | NA | 8 | NA |
| **REC opinion** | | | | | | |
| Favourable | 942 | 98 | 798 | 98 | 1740 | 98 |
| Favourable | 200 | 21 | 217 | 27 | 417 | 23 |
| Favourable (extra info) | 742 | 77 | 581 | 72 | 1323 | 75 |
| Unfavourable | 21 | 2 | 14 | 2 | 35 | 2 |

*Not mutually exclusive.

CTIMP, clinical trials of investigational medicinal product; RCT, randomised controlled trial; REC, Research Ethics Committee; SmPC, summary of product characteristics.

the relevant questions on the form, or instances where these words were used near each other, where 'near' was defined as being within three words. For adaptive designs, we employed the US Food and Drug Administration (FDA) definition, namely 'a study that includes a prospectively planned opportunity for modification'.[17] We extracted the sentence containing the target phrase as well as one sentence before and after into a separate document. These textural extracts were then reviewed by the authors to identify true and false positives.

The data extracted from the separate XML files were collated into one dataset and descriptive analyses of this dataset were done with Stata.

## RESULTS

We received (30 June 2016) the XML files for 1814 application records submitted for ethical review during the period specified, and extracted by HRA from the IRAS system. Of these, 1659 (92%) were from 2015, 154 (9%) from 2014 and 1 was from 2016. Three records were corrupted and could not be processed. In discussion with HRA, we discarded some records to remove duplicates. We kept one of four entries for the WHEAT trial which had been sent to multiple committees as part of research on consistency of REC opinions.[18] We discarded one of two entries for a further study, which was initially given a favourable opinion and then submitted additionally to a specialist REC in Scotland (again, favourably). Thirty-two further records were discarded that initially had an unfavourable opinion before re-submission as a (near) identical, separate application for review. We discarded the first applications and kept the re-submissions, regardless of the subsequent review's outcome. Therefore, our dataset included 1775 studies.

The filter questions switched off, for some applications, questions in which we were interested. For example, trial phase was only recorded for CTIMPs and the use of data monitoring committees was infrequently recorded. Furthermore, the use of blinding was not sufficiently well captured in the system to allow us to present reliable data.

A total of 963/1775 (54%) of the applications were stated as being RCTs. Table 1 describes the disease setting, research method, CTIMP and REC opinion by whether they were an RCT. The most common research area, using the categorisation of the IRAS system, was cancer, followed by cardiovascular, musculoskeletal and respiratory diseases; around one-third of records specified more than one area. This is broadly consistent with previous reviews.[19][20] Around one-fifth of RCTs were pilot studies; one-third of records were employing more than one method. The REC gave a favourable opinion to the vast majority of applications (1740/1775, 98%), but required additional information for most of these 1323/1740 (75%).

Over half of the RCTs (515/963, 53%) were CTIMPs, of which 315/515 (61%) were CTIMPs of unlicensed products, 146/515 (28%) licensed CTIMPs in a new setting and 66/515 (13%) CTIMPs used according to their summary of product characteristics (SmPC); the categories were not mutually exclusive. Over a third (284/812, 35%) of the other studies (not RCTs) were also CTIMPs.

Table 2 describes the plans in design and dissemination. Most RCTs (895/963, 93%) were not recorded as being preceded by a formal systematic review of the literature. Only 15 (2%) RCTs were detectably designed under a Bayesian framework; 20 (2%) were detectable as having an adaptive design. There were 34 cluster randomised and 5 stepped wedge trials, overall. Six hundred fourteen (63%) RCTs were requesting new biological samples; 120 (12%) RCTs were seeking access to previous biological samples.

The large majority of research (1667/1775, 94%) was planned for dissemination in the peer-reviewed literature and most (1585/1775, 89%) were planned for conference presentations. A total of 475/963 (49%) RCTs and 352/812 (43%) other studies were planned for submission

| Table 2 | Issues in design and dissemination in all entries | | | | | | |
|---|---|---|---|---|---|---|---|
| | **RCT** | | **Other** | | **Total** | | |
| **Characteristic** | **N** | **%** | **N** | **%** | **N** | **%** | |
| N | 963 | | 812 | | 1775 | | |
| **Review of data as part of development** | | | | | | | |
| Neither | 895 | 93 | 759 | 93 | 1654 | 93 | |
| Systematic review only | 38 | 4 | 32 | 4 | 70 | 4 | |
| Meta-analysis only | 18 | 2 | 13 | 2 | 31 | 2 | |
| Both | 12 | 1 | 8 | 1 | 20 | 1 | |
| **Design characteristics** | | | | | | | |
| Neither | 926 | 96 | 778 | 96 | 1704 | 1 | |
| Adaptive design | 20 | 2 | 9 | 1 | 29 | 2 | |
| Bayes design | 15 | 2 | 24 | 3 | 39 | 2 | |
| Both | 2 | 0 | 1 | 0 | 3 | 0 | |
| Cluster randomised | 26 | 3 | 8 | 1 | 34 | 2 | |
| Stepped wedge | 3 | 0 | 2 | 0 | 5 | 0 | |
| **Sample collection*** | | | | | | | |
| Taking new samples | 614 | 64 | 422 | 52 | 1036 | 58 | |
| Accessing stored samples | 120 | 12 | 110 | 14 | 230 | 13 | |
| **Plans for dissemination†** | | | | | | | |
| Peer-reviewed scientific journal | 912 | 95 | 755 | 93 | 1667 | 94 | |
| Conference presentation | 869 | 90 | 716 | 8 | 1585 | 89 | |
| Internal report | 641 | 67 | 526 | 65 | 1167 | 66 | |
| Publication on website | 536 | 56 | 367 | 45 | 903 | 51 | |
| Submission to regulatory authority | 475 | 49 | 352 | 43 | 827 | 47 | |
| Other publication | 239 | 25 | 178 | 22 | 417 | 23 | |
| Access to raw data | 191 | 20 | 133 | 16 | 114 | 6 | |
| Other | 0 | 0 | 3 | 0 | 324 | 18 | |
| No plans to report or disseminate | | | | | 3 | 0 | |

*Not asked for non-regulatory RCTs.
†Not mutually exclusive.
RCT, randomised controlled trial.

**Table 3** Reported PPI

| Characteristic | RCT | | Other | | Total | |
|---|---|---|---|---|---|---|
| | N | % | N | % | N | % |
| N | 963 | | 812 | | 1775 | |
| **Areas of PPI activity*** | | | | | | |
| Design of the research | 403 | 42 | 323 | 40 | 726 | 41 |
| Dissemination of findings | 375 | 39 | 284 | 35 | 659 | 37 |
| Undertaking the research | 242 | 25 | 219 | 27 | 461 | 26 |
| Management of the research | 225 | 23 | 129 | 16 | 354 | 20 |
| Analysis of results | 72 | 7 | 75 | 9 | 147 | 8 |
| **Number of areas of PPI activity** | | | | | | |
| None | 389 | 40 | 302 | 37 | 691 | 39 |
| 1 | 205 | 21 | 209 | 26 | 414 | 23 |
| 2 | 139 | 14 | 117 | 14 | 256 | 14 |
| 3 | 125 | 13 | 93 | 11 | 218 | 12 |
| 4 | 66 | 7 | 37 | 5 | 103 | 6 |
| All | 39 | 4 | 32 | 4 | 71 | 4 |

*Not mutually exclusive.
PPI, patient and public involvement.

to regulatory authorities. A small number, 114/1775 (6%), were planning to offer raw data to external applicants as a key form of dissemination.

Table 3 notes the reported patients and public involvement (PPI) using the categories specified by IRAS. A total of 726/1775 (41%) of studies claim PPI in the design and 659/1775 (37%) claim planned PPI in dissemination of the findings. PPI engagement in undertaking and management of the research was less common, and few studies (147/1775, 8%) involved planned PPI in analysis. Around one-third of studies involve PPI in two or more of these IRAS-defined areas.

Table 4 shows within CTIMPs the similarities and differences across the phase of research. Trial phase was asked as a series of separate yes/no questions for phase I, II, III and IV. These were not mutually exclusive. For the purpose of summarising, we selected the highest level if more than one option was selected. Early phase trials were more likely to be designed within a Bayesian framework; phase III trials most often reported the use of a placebo and the use of an Independent Data Monitoring Committee; phase IV trials were least likely to be submitted to a regulatory authority or put on a website, but most often reported PPI in all five broad areas.

Table 5 shows the outcome of the manual review of the extracted free-text fields. Accuracy, defined as number of true positives divided by overall total ranged from 43% for adaptive designs to 100% for stepped wedge. The false positives related to studies described as 'phase 1/2' or 'phase 2/3', but with no evidence of an adaptive step, for example, phase 1/2 was often used to describe a classical pharmacokinetic study in patients rather than healthy volunteers; the database did not contain healthy

volunteer studies. For 'systematic review' and 'meta-analysis', the search often picked up references to previous studies and not the planned study.

## DISCUSSION

We achieved our primary feasibility aim of negotiating access to the centralised UK approvals system and devising a way of extracting information from a series of separate XML files. We anticipate that the programmes we developed could extract further pertinent points of information with minimal manual involvement. We were also able to achieve our primary descriptive aim of systematically reporting on the state of clinical research in the UK over around 1 year, focused on 2015.

We found that nearly 1000 RCTs were submitted for approval in the UK during that period and >800 other studies. A key message is the volume of research activity in the UK, with the country demonstrably research active. The majority of applications received a favourable opinion but most required further information first, suggesting that applications could be better completed, saving time and effort for all parties.

It was notable that the reported use of adaptive designs was low and was not markedly different from earlier estimates.[19 20] This is disappointing, as many members of this design family have scope to reduce time to answer and/or reduce the average cost per answer, particularly by moving away early from treatment approaches that are not likely to improve outcomes sufficiently for patients.[21] However, Bayesian approaches have penetrated 10% of early phase CTIMPs. The reported use of IDMCs for phase III CTIMPs was, perhaps, low at 58%, but this

**Table 4** Characteristics of the CTIMPs by trial phase

| Trial phase* | Ph I | | Ph II | | Ph III | | Ph IV | | Total | |
|---|---|---|---|---|---|---|---|---|---|---|
| **Characteristic** | **N** | **%** | **N** | **%** | **N** | **%** | **N** | **%** | **N** | **%** |
| All entries, N | 99 | | 287 | | 324 | | 98 | | 808 | |
| **Design characteristics** | | | | | | | | | | |
| Neither | 86 | 89 | 261 | 93 | 303 | 94 | 90 | 94 | 740 | 93 |
| Adaptive | 1 | 1 | 7 | 2 | 13 | 4 | 3 | 3 | 24 | 3 |
| Bayes | 10 | 10 | 12 | 4 | 5 | 2 | 2 | 2 | 29 | 4 |
| Both | 0 | 0 | 2 | 1 | 0 | 0 | 1 | 1 | 3 | 0 |
| Cluster randomised | 0 | 0 | 0 | 0 | 0 | 0 | 0 | 0 | 0 | 0 |
| Stepped wedge | 0 | 0 | 0 | 0 | 0 | 0 | 0 | 0 | 0 | 0 |
| Involves placebo | 21 | 22 | 123 | 44 | 163 | 51 | 25 | 26 | 332 | 42 |
| **Independent data monitoring committee** | | | | | | | | | | |
| No | 40 | 82 | 84 | 59 | 61 | 42 | 46 | 78 | 231 | 58 |
| Yes | 9 | 18 | 59 | 41 | 84 | 58 | 13 | 22 | 165 | 42 |
| *Missing* | *8* | | *139* | | *176* | | *37* | | *400* | |
| **Methods of dissemination** | | | | | | | | | | |
| Peer-reviewed scientific journal | 86 | 89 | 265 | 94 | 290 | 90 | 93 | 97 | 86 | 89 |
| Submission to regulatory authority | 85 | 88 | 215 | 76 | 272 | 85 | 48 | 50 | 85 | 88 |
| Conference presentation | 82 | 85 | 253 | 90 | 271 | 84 | 89 | 93 | 82 | 85 |
| Internal report | 71 | 73 | 212 | 75 | 259 | 81 | 60 | 63 | 71 | 73 |
| Publication on website | 56 | 58 | 161 | 57 | 203 | 63 | 45 | 47 | 56 | 58 |
| Other publication | 26 | 27 | 76 | 27 | 92 | 29 | 17 | 18 | 26 | 27 |
| Access to raw data | 8 | 8 | 12 | 4 | 25 | 8 | 8 | 8 | 8 | 8 |
| Other | 12 | 12 | 47 | 17 | 48 | 15 | 18 | 19 | 12 | 12 |
| No plans to report or dissemination | 0 | 0 | 0 | 0 | 2 | 1 | 0 | 0 | 0 | 0 |
| **Areas of PPI activity†** | | | | | | | | | | |
| Design of the research | 26 | 27 | 82 | 29 | 39 | 12 | 50 | 52 | 26 | 27 |
| Management of the research | 10 | 10 | 34 | 12 | 28 | 9 | 31 | 32 | 10 | 10 |
| Undertaking the research | 14 | 14 | 47 | 17 | 36 | 11 | 29 | 30 | 14 | 14 |
| Analysis of results | 1 | 1 | 15 | 5 | 7 | 2 | 12 | 13 | 1 | 1 |
| Dissemination of findings | 20 | 21 | 74 | 26 | 64 | 20 | 44 | 46 | 20 | 21 |
| None of the above | 53 | 55 | 149 | 53 | 223 | 69 | 35 | 36 | 460 | 58 |
| **REC opinion** | | | | | | | | | | |
| Favourable | 97 | 98 | 277 | 97 | 318 | 98 | 93 | 95 | 785 | 97 |
| Favourable | 14 | 14 | 51 | 18 | 40 | 12 | 15 | 15 | 120 | 15 |
| Favourable (after extra info) | 83 | 84 | 226 | 79 | 278 | 86 | 78 | 80 | 665 | 82 |
| Unfavourable | 0 | 0 | 5 | 2 | 3 | 1 | 3 | 3 | 11 | 1 |

*Trial phase calculated.
†Not mutually exclusive.
CTIMP, clinical trials of investigational medicinal product; PPI, patient and public involvement; REC, Research Ethics Committee.

is a substantial increase over the prevalence in trials published in high impact-factor journals in 1990 and 2000.[22] Involvement of patients and the public in various aspects of research is increasingly seen as important, but reported rates of engagement were quite low: more than half of studies had no PPI involvement.[23] Cursory review of the free-text fields associated with these PPI categories suggests that some applicants may have been a little generous in choosing to select a particular category, suggesting that the numbers we report may be overestimates. Other comments suggest that PPI is not needed, perhaps reflecting that the value of PPI is yet to

**Table 5** Review of possible search terms

| Design element Search term | Performance of search term* | | | | | |
| | At sentence level | | | At study level | | |
| | True positive | False positive | Accuracy | True positive | False positive | Accuracy |
|---|---|---|---|---|---|---|
| **Systematic review** | | | | | | |
| Review (near) systematically | 2 | 0 | 100 | 2 | 0 | 100 |
| Evidence (near) review | 4 | 1 | 80 | 4 | 1 | 80 |
| Systematic (near) review | 98 | 39 | 72 | 58 | 23 | 72 |
| Literature (near) review | 29 | 13 | 69 | 26 | 12 | 68 |
| *Evidence-based review* | *0* | *0* | *NA* | *0* | *0* | *NA* |
| Summary*† | 133 | 53 | 72 | 90 | 36 | 71 |
| **Meta-analysis** | | | | | | |
| Pooled analysis | 2 | 0 | 100 | 2 | 0 | 100 |
| Meta-analysis | 77 | 31 | 71 | 49 | 19 | 72 |
| *Integrated analysis* | *0* | *0* | *NA* | *0* | *0* | *NA* |
| Summary*† | 79 | 31 | 72 | 51 | 19 | 73 |
| **Adaptive design** | | | | | | |
| Adaptive design | 19 | 0 | 100 | 13 | 0 | 100 |
| Adaptive randomisation | 8 | 0 | 100 | 5 | 0 | 100 |
| Continual reassessment | 2 | 0 | 100 | 1 | 0 | 100 |
| Sample size re-estimation | 15 | 0 | 100 | 9 | 0 | 100 |
| Seamless design | 1 | 0 | 100 | 1 | 0 | 100 |
| Phase I/II | 4 | 43 | 9 | 3 | 37 | 8 |
| Phase II/III | 2 | 25 | 7 | 2 | 23 | 8 |
| *CRM design* | *0* | *0* | *NA* | *0* | *0* | *NA* |
| *Drop-the-loser* | *0* | *0* | *NA* | *0* | *0* | *NA* |
| *MAMS design* | *0* | *0* | *NA* | *0* | *0* | *NA* |
| *Multiarm multistage* | *0* | *0* | *NA* | *0* | *0* | *NA* |
| *Pick-the-winner* | *0* | *0* | *NA* | *0* | *0* | *NA* |
| Summary*† | 51 | 68 | 43 | 31 | 60 | 34 |
| **Bayes** | | | | | | |
| Bayes | 83 | 2 | 98 | 42 | 1 | 98 |
| Summary*† | 83 | 2 | 98 | 42 | 1 | 98 |
| **Cluster RCT** | | | | | | |
| Cluster RCT | 26 | 0 | 100 | 11 | 0 | 100 |
| Control cluster | 6 | 0 | 100 | 4 | 0 | 100 |
| Randomised cluster | 2 | 0 | 100 | 1 | 0 | 100 |
| Cluster random | 46 | 1 | 98 | 30 | 1 | 97 |
| Summary*† | 80 | 1 | 99 | 34 | 1 | 97 |
| **Stepped wedge** | | | | | | |
| Step (near) wedge | 1 | 0 | 100 | 1 | 0 | 100 |
| Stepped (near) wedge | 16 | 0 | 100 | 5 | 0 | 100 |
| Summary*† | 17 | 0 | 100 | 5 | 0 | 100 |

NB The table reports on whether each search term accurately identified a specific design element of the study in selected free-text fields. All variations of the spelling, capitalisation, hyphenation and so on were covered in the search terms used.

*The left side of the table separately considers each sentence which matched a search term accurately identified the design property. There were 613 such sentences. The Summary row in each subsection is the sum of the constituent search terms.

†The right side of the table separately considers whether any sentence in a study matching a search term accurately identified the design property. Some studies matched a search term repeatedly, therefore the numbers are smaller. The Summary row in each subsection reflects a search across all of its constituent search terms. Note that some studies matched more than one search term in a subsection, so the Summary row may not be the sum of the rows above for example, a study that uses both 'cluster random' and 'control cluster' would appear only once in the Summary row, and would appear separately in the 'cluster random' and 'control cluster' rows.

CRM, continual reassessment method; MAMS, multi-arm multi-stage; RCT, randomised controlled trial.

be recognised by some researchers.[3] Further research in this area is required.

Most but not all studies were planned for publication; there is still some way to go on this transparency aim. That raw data would be made available for some studies is encouraging and we hope that this will increase with time for appropriate projects from qualified researchers.[24]

Extracting the information from the HRA system was more effortful than expected. Moreover, important characteristics of clinical trials like blinding could not be analysed and certain search terms clearly had limitations. The issue of blinding is particularly important, as it is one of the desirable characteristics in generating reliable evidence from clinical trials.[11] However, now that we have completed this feasibility step, it is possible for future research projects to access an expanded dataset and work with the HRA to organise the database in a way that will facilitate its analysis. Access to several years of data would allow researchers, as well as the HRA itself, to examine the progress of initiatives to improve the quality of clinical research, such as those designed to encourage PPI and publication of clinical trial results.[13 15] We would also encourage the HRA to adopt standard definitions to describe the characteristics of clinical trials, as done in the USA.[25] Confusion over the definition of terms greatly complicates efforts to automatically analyse clinical research.

Our pilot work has some limitations. We had no straightforward way of aligning the questions in the PDF document with the information stored in the XML document. Perhaps if we had negotiated access to the backend software used by IRAS to generate these PDF forms, we may have been able to do this automatically; instead, we needed to reverse-engineer the IRAS system in order to figure out which XML tags were responsible for creating the content we wanted to study. Not all of the applications had been created using the same version of the IRAS tool.[16] Although a large majority of the documents were created using the latest version of the tool, eight different versions were present in the dataset we received. We did not receive any guidance on the ways in which one version of the tool differed from another version, and we only had access to the latest version. We made efforts to determine the specific situations in which questions were not asked rather than asked but not answered, in order to make this distinction ourselves, though ideally this could be indicated directly in the XML dataset. The labelling of missing answers and inactivated questions was not consistent. There is scope for applicants to misunderstand the questions and misrepresent the data. For example, three trials were noted as using RCT methodology but answered 'no' to the question: 'Will participants be allocated to groups at random?' We checked the data and found these were not actually RCTs; we included them with the 'other studies'. Likely this error by the applicants activated or de-activated some questions inappropriately. In this example, the RCT method question actually asked about randomisation of individuals and these three

applicants who submitted cluster randomised trials had answered this question inconsistently. Finally, the search terms chosen for the free-text fields were not exhaustive and because it was unfeasible in a pilot to review all applications, the sensitivity and specificity of the free-text extraction could not be calculated.

Some interesting information was held only in the free-text fields. We used standard methods to extract this information using automation supplemented by manual checking. There is a wealth of free-text information, which through the application of text-mining techniques could provide incredibly valuable insights into the characteristics of health research. For this pilot study, we did not have the resources to review all of the questions and we particularly could not spend time reading free-text fields. Therefore, we felt obliged not to correct categorical questions if we noticed they were contrasted by free-text; however, a fully-resourced study could do this. For example, we noticed at least one entry, which ticked boxes for PPI engagement and expanded on this by claiming that, as doctors, they knew what patients wanted and did not need to trouble patients for their time on these aspects.

In conclusion, we have demonstrated that anonymised data from the HRA's research system are accessible and can be queried for information. We strongly encourage the development of fully resourced collaborative projects to delve more deeply into this data. We believe that it is imperative that the characteristics of clinical research in the UK are understood, as these underpin clinical guidance. Finally, there are many ongoing initiatives to improve the quality of clinical research, but only by fully understanding the state of, and changes to, the research profile of the UK, can we appreciate their impact.

**Acknowledgements** The authors would like to thank Hazel Gage, Amanda Hunn, Janet Messer—UK Health Research Authority for providing data, engaging in discussion and commenting on the draft manuscript and Professor Dr Ulrich Mansmann, IBE, Munich, Germany for commenting on the draft manuscript.

**Contributors** TPC and MRS: conceptualised the study, designed the study, collected the data, analysed and interpreted the data, wrote critical sections of manuscript, and reviewed and approved final manuscript. RHW: designed the study, collected the data, analysed and interpreted the data, wrote critical sections of manuscript, and reviewed and approved final manuscript.

**Funding** This study was funded by Medical Research Council (grant no: MC_UU_12023/24).

**Competing interests** TPC: worked as a consultant for the clinical research organisation ICON in the previous 3 years.

**Patient consent** Not required.

**Provenance and peer review** Not commissioned; externally peer reviewed.

**Data sharing statement** The data used in this analysis were obtained from the Health Research Authority under a confidentiality agreement and is not under the researcher's gift to share. Applications for these data or expanded data must involve the HRA. Access to the processing code is available through the MRC Clinical Trials Unit at UCL's usual Data Release Request approach.

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
