## [Reviewer comments · BMJ Open]

ARTICLE DETAILS

TITLE (PROVISIONAL)	A cross-sectional analysis of UK research studies in 2015: results from a scoping project with the UK Health Research Authority
AUTHORS	Clark, Tim ; Wicentowski, Richard; Sydes, Matthew

VERSION 1 – REVIEW

REVIEWER	Mark Elwood University of Auckland, New Zealand
REVIEW RETURNED	26-Feb-2018

GENERAL COMMENTS	Specific comments. Abstract. Can we explain 'XML' files for the general reader? The title and opening statement talk about 'trials'; 963 were randomised trials, implying that the rest were non-randomised trials? Were these other studies actually trials - intervention studies - or do they include observational studies etc? 'A favourable opinion' is unclear. Do you mean there were approved by the ethics review? Strengths and limitations. Several unexplained abbreviations. Introduction. Opening paragraph talks about clinical trials as the gold standard; but this applies to randomised trials. Throughout this paper, 'trial' is used loosely and it's not clear what it means. Sometimes it seems to include all studies. Similar in the second paragraph is there mandatory registration of trials, or only of randomised trials? End of introduction: 'phase 1 commercial trials' were excluded: were these easy to identify?
---

	Methods. General readers will not be computer experts, so the last sentence in the first paragraph of methods is likely to be incomprehensible. Can we explain it better? In the second paragraph, 'healthy volunteer studies' are also excluded; not noted in the introduction or elsewhere. How were these identified? Results 'the period specified' is not in fact specified; in the Methods it's 2014 to 2016, but the studies are nearly all from 2015. Do they represent all the applications in 2015? Why the small number in other years, and would it be better to exclude them? The first Table mentioned in the results is Table 2. Table 2 has two columns for RCT and other, but also several rows relating to RCTs, and it not clear which data is represented. In the RCT column, there are observational studies, studies with no randomisation, case reviews, and several other designs which would seem incompatible with the RCT label. Under dissemination, was feedback of results to the participants included? The text says Table five relates to clinical trials, but it relates only to CTIMPs. Table 1, mentioned last, needs to be redone. It appears to be a comparison of manual review versus some other type of review of the data. It's not clear in the methods what was done. Such cross-classification gives four categories, but only two are shown. Accuracy is defined in the text as true positives divided by the overall total, but appears in the table to be true positives divided by all positives, which may be sensitivity, but is not overall accuracy. There is nothing to justify calling the results of one review the 'true' results, unless there is further evidence to support that. I would like to see much more description of what the studies were which were not RCT's. Discussion. The paper is basically the descriptive results of a computer-based extraction system, and can be accepted as that, except that many aspects (as noted above) are not very clear.
--	---

	The interpretation in terms of qualities and standards for clinical trials is more difficult. For instance it stated that most applications needed further information before they were approved, and the authors suggest that applications could be better competed. But the difficulty may be that the requirements of these committees are unclear or do not match the application forms. The authors note that reported rates of involvement of patients and the public were quite low. It strikes me that they are high, but I wonder what a positive response actually means? Getting meaningful wide involvement in the design and conduct of studies is a very demanding task. The authors note in this paragraph that the response to the question may be 'generous', but further comment would be useful. The plea to encourage the HRA to adopt standard definitions seems an obvious essential step. The other general issue is that the forms presumably are not designed for this type of analysis, and if they are to be used in such analyses routinely, they could be much more usefully designed. The fact that different versions of forms have been used without adequate documentation says a lot. Some of the discussion is mainly about the limitations of the study methods, which will not be of general interest, and could be shortened considerably. Summary. It needs to be more carefully written. Terms like 'trials' are used very loosely (including in the title). More needs to be said about the large number of studies which were not RCTs. Table 1 on the analysis behind it is not well described or interpreted, and needs revision.
--	---

REVIEWER	Sean Coady National Heart, Lung, and Blood Institute; United States
REVIEW RETURNED	28-Mar-2018

GENERAL COMMENTS	From a funding perspective there is a clear interest to fully understand the nature of research itself. This is particularly relevant for publicly funded research. There is a reasonable expectation on the part of the public that research monies are being used wisely, and therefore research about research is a topic of great interest especially in an era of increasing capacity to generate, store and analyze data from databases primarily used for administrative purposes.
--

	The article “A cross-sectional study of UK trials in 2015: results from a scoping project with the UK Health Research Authority” is a welcomed analysis of a potential approach to gain insights into the nature of research. Though fairly narrow in scope the article presents an approach that could be scaled up to answer a range of questions related to human subjects research. Being unfamiliar with the process for Ethical approval in the UK, some added material on the HRA may be helpful to non-UK readers. For example, are all trials submitted to HRA for Ethical Approval or are some trials eligible for local approval at the institution of the investigator? If local approvals are possible, what determines which studies qualify for local approval? Are continuing reviews submitted to HRA? Do closeout activities have to be submitted to HRA? Additional descriptions of the HRA in the life cycle of a trial would help readers understand the potential of HRA data in understanding how such data could further research about research. For example, a persistent problem for funders is planned versus actual enrollment. If HRA submissions occur throughout a trial’s life cycle, then such data could lead to insights on when and why recruitment might be problematic. Specific comments:  1) While it is understood that determining sensitivity and specificity across all records included in the study would be problematic, examining sensitivity and specificity of the text mining methodology in a sample of records would significantly enhance the manuscript. 2) It was noted in the text that the search terms “were not exhaustive”. Was any methodology employed to determine the search terms that were used? 3) The sentence beginning on line 177 (page 11) is unclear. It almost appears that the start of the paragraph was cut off somehow. 4) Table 1. The table indicates many more true positives than indicated in Table 3. Is this due to multiple true-positives per application? 5) Table 2. A footnote is indicated for Therapeutic Area; however, I could not locate the footnote text. 6) Table 3. A footnote is included at the bottom of the table; however, it is unclear to what the footnote should be attached. 7) Table 4. A footnote is indicated for Areas of PPI activity; however, I could not locate the footnote text.
--	--

VERSION 1 – AUTHOR RESPONSE

Comment	Response
Reviewer 1 -- Mark Elwood (University of Auckland)	
Specific comments.	
Abstract	
Can we explain ‘XML’ files for the general reader?	We have added an explanation in the Methods: “XML is a commonly-used file format that stores structured data in a plain text format.”
The title and opening statement talk about ‘trials’; 963 were randomised trials, implying that the rest were non-randomised trials? Were these other studies actually trials - intervention studies - or do they include observational	We have used the term “study” throughout to cover all types of clinical studies.

studies etc?	
'A favourable opinion' is unclear. Do you mean there were approved by the ethics review?	This is the preferred term of the HRA. They do not approve, they give a "favourable opinion".
Strengths and limitations	
Several unexplained abbreviations.	The abbreviations IRAS, HRA and HARP are all explained when they are first used in the abstract. We will look to the editorial team to advise whether we have missed any further abbreviations.
Introduction	
Opening paragraph talks about clinical trials as the gold standard; but this applies to randomised trials. Throughout this paper, 'trial' is used loosely and it's not clear what it means. Sometimes it seems to include all studies.	The reviewer is correct. We were particularly interested in trials, but we have a broad range of clinical studies. Therefore, we have updated to inclusively use "study" throughout.
Similar in the second paragraph is there mandatory registration of trials, or only of randomised trials?	There is mandatory registration of all clinical studies. ClinicalTrials.gov for example, includes both interventional and observational studies. The paragraph has been revised accordingly.
End of introduction: 'phase 1 commercial trials' were excluded: were these easy to identify?	These were filtered by HRA before we received the dataset so we have not seen these studies.

Comment	Response
Methods	
General readers will not be computer experts, so the last sentence in the first paragraph of methods is likely to be incomprehensible. Can we explain it better?	We agree this could be clearer. We have introduced the description of XML files, and split the sentence into three simpler sentences.
In the second paragraph, 'healthy volunteer studies' are also excluded; not noted in the introduction or elsewhere. How were these identified?	As stated above, the HRA excluded Phase I Healthy Volunteer studies, as these are commercially sensitive. The paragraph has been revised accordingly.
Results	
'the period specified' is not in fact specified; in the Methods it's 2014 to 2016, but the studies are nearly all from 2015. Do they represent all the applications in 2015? Why the small number in other years, and would it be better to exclude them?	We asked for studies from 2015. There were some stray studies that HRA provided from late 2014 and early 2016. After deliberation, we chose to include all of the studies they provided to us, as this is pilot work and we felt that the enhanced sample was more useful.
The first Table mentioned in the results is Table 2.	Table 1 was mentioned in the Methods as a way to direct the reader towards the search terms we used. However, we agree that this could be confusing, so we have made more mention of examples in the

	Methods so that the table needn't serve this purpose. This allows us to first mention all of the tables in the Results so we have reordered the table numbers to the more preferable ordering.
Table 2 has two columns for RCT and other, but also several rows relating to RCTs, and it not clear which data is represented. In the RCT column, there are observational studies, studies with no randomisation, case reviews, and several other designs which would seem incompatible with the RCT label.	The research methods section in Table 2 is not mutually exclusive; for example, an RCT could include a pilot/feasibility stage and a questionnaire. Two studies claimed to be randomised controlled trials but did not appear to be when we looked at the records, so we assume that a wrong option had been selected. (This would have implications for those researchers about which form questions they were then asked...)
Under dissemination, was feedback of results to the participants included?	This was not clear from the documentation reviewed.
The text says Table five relates to clinical trials, but it relates only to CTIMPs.	We have corrected this labelling; the phase question was only asked about CTIMPs.
Table 1, mentioned last, needs to be redone. It appears to be a comparison of manual review versus some other type of review of the data. It's not clear in the methods what was done. Such cross-classification gives four categories, but only two are shown. Accuracy is defined in the text as true positives divided by the overall total, but appears in the table to be true positives divided by all positives, which may be sensitivity, but is not overall accuracy.	We have changed Table 1 extensively. As part of this, accuracy is now consistently defined as true positives out of all positives. Thank you for spotting this typo.
There is nothing to justify calling the results of one review the 'true' results, unless there is further evidence to support that.	We considered the results of the manual review to be the "true result", as this was performed by an individual with the knowledge and experience to accurately identify the characteristic in question.

Comment	Response
I would like to see much more description of what the studies were which were not RCT's.	We wished to focus on RCTs, ideally. We agree that expansion would be nice. We would prefer to work on this in the full study, if we are able to secure the necessary funding and collaboration.
Discussion	
The paper is basically the descriptive results of a computer-based extraction system, and can be accepted as that, except that many aspects (as noted above) are not very clear. The interpretation in terms of qualities and standards for clinical trials is more difficult. For instance it stated that most applications needed further information before they were approved, and the authors	The paper is a description of the dataset held by HRA, descriptions that are not widely circulated. The extraction system allowed us to consider these data and sets us up for future, larger (funded) research. We agree with the concern raised by the reviewer. If there is a disconnect between what committees want and what committees are being provided, this

suggest that applications could be better competed. But the difficulty may be that the requirements of these committees are unclear or do not match the application forms.	is a problem with the system. Highlighting it is a step to resolution, a resolution HRA needs to put in place.
The authors note that reported rates of involvement of patients and the public were quite low. It strikes me that they are high, but I wonder what a positive response actually means? Getting meaningful wide involvement in the design and conduct of studies is a very demanding task. The authors note in this paragraph that the response to the question may be 'generous', but further comment would be useful.	The UK has been working hard to improve PPI across studies, so the number struck us as lower than might be expected, but perhaps higher than many other places have now or historically. There is some further detail about what a positive response is reported to mean which we would dig into in a full study.
The plea to encourage the HRA to adopt standard definitions seems an obvious essential step. The other general issue is that the forms presumably are not designed for this type of analysis, and if they are to be used in such analyses routinely, they could be much more usefully designed. The fact that different versions of forms have been used without adequate documentation says a lot.	We agree.
Some of the discussion is mainly about the limitations of the study methods, which will not be of general interest, and could be shortened considerably.	We think the space given over to the limitations is proportionate, about two fifths of the space, given that one key aim is about feasibility. We wanted to set out clearly the challenges we faced so that anyone who takes this work forward, whether we or others, goes in aware. However, we will seek further editorial guidance.
Summary	
It need to it needs to be more carefully written. Terms like 'trials' are used very loosely (including in the title). More needs to be said about the large number of studies which were not RCTs.	We have used the term "study" throughout to cover all types of clinical studies. We wished to focus on RCTs, ideally. We agree that expansion would be nice. We would prefer to work on this in the full study, if we are able to secure the necessary funding and collaboration.
Table 1 on the analysis behind it is not well described or interpreted, and needs revised.	Table 1 has now been extensively revised. We are now clearer that the previous data related to sentences that had met the triggers. We have also added whether there was any trigger for a given term in a study in a new set of columns.

Comment	Response
Reviewer 2 -- Sean Cody (NHLBI)	
From a funding perspective there is a clear interest to fully understand the nature of research itself. This is particularly relevant for publicly funded research. There is a reasonable expectation on the part of the public that research monies	No response required.

are being used wisely, and therefore research about research is a topic of great interest especially in an era of increasing capacity to generate, store and analyze data from databases primarily used for administrative purposes.	
The article “A cross-sectional study of UK trials in 2015: results from a scoping project with the UK Health Research Authority” is a welcomed analysis of a potential approach to gain insights into the nature of research. Though fairly narrow in scope the article presents an approach that could be scaled up to answer a range of questions related to human subjects research.	No response required.
Being unfamiliar with the process for Ethical approval in the UK, some added material on the HRA may be helpful to non-UK readers. For example, are all trials submitted to HRA for Ethical Approval or are some trials eligible for local approval at the institution of the investigator? If local approvals are possible, what determines which studies qualify for local approval? Are continuing reviews submitted to HRA? Do closeout activities have to be submitted to HRA? Additional descriptions of the HRA in the life cycle of a trial would help readers understand the potential of HRA data in understanding how such data could further research about research. For example, a persistent problem for funders is planned versus actual enrollment. If HRA submissions occur throughout a trial’s life cycle, then such data could lead to insights on when and why recruitment might be problematic.	The HRA oversees the ethical review of research conducted within the National Health Service (NHS). Research conducted outside of the NHS does not require HRA approval, but may still need ethical approval from a local REC. The text has been revised to explain this.
Specific comments:	
1) While it is understood that determining sensitivity and specificity across all records included in the study would be problematic, examining sensitivity and specificity of the text mining methodology in a sample of records would significantly enhance the manuscript.	We understand the comment of the reviewer, but we are not sure that this would add significantly to the manuscript, as single terms were not sensitive enough to identify certain characteristics, such as adaptive studies. This conclusion is not likely to be altered by reviewing a sample of records. This would be explored in more detail in a full study, with funding.
2) It was noted in the text that the search terms “were not exhaustive”. Was any methodology employed to determine the search terms that were used?	We discussed and agreed commonly used key terms. We anticipate that there are less commonly used terms that some people prefer. This would be explored in more detail in a full study, with funding.
3) The sentence beginning on line 177 (page 11) is unclear. It almost appears that the start of the paragraph was cut off somehow.	The sentence appears to be complete in our version of the submitted manuscript: “The filter questions switched off, for some applications, questions in which we were interested.” We have left this unchanged in the revised version.

Comment	Response
4) Table 1. The table indicates many more true positives than indicated in Table 3. Is this due to multiple true-positives per application?	Table 1 has now been extensively revised. We are now clearer that the previous data related to sentences that had met the triggers. We have also added whether there was any trigger for a given term in a study in a new set of columns.
5) Table 2. A footnote is indicated for Therapeutic Area; however, I could not locate the footnote text.	We have checked and updated the footnotes. The previously-missing one was to highlight that certain terms were not mutually exclusive.
6) Table 3. A footnote is included at the bottom of the table; however, it is unclear to what the footnote should be attached.	We have checked and updated the footnotes. The previously-missing one was to highlight that certain terms were not mutually exclusive.
7) Table 4. A footnote is indicated for Areas of PPI activity; however, I could not locate the footnote text.	We have checked and updated the footnotes. The previously-missing one was to highlight that certain terms were not mutually exclusive.

VERSION 2 – REVIEW

REVIEWER	Sean Coady National Heart, Lung, and Blood Institute, USA
REVIEW RETURNED	12-Jun-2018

GENERAL COMMENTS	Line 56. Reference to "gold standard" may need to be removed or revised since the relevant sentence was broadened from RCTs to clinical studies Line 70, suggesting rephrasing to "where registration for clinical trials is mandatory" Line 90. Suggest changing clinical trials to clinical studies Abstract and line 110. Suggest considering rephrasing the timeframe to: "applications for clinical trials submitted for review primarily in 2015" (note line 110 indicates clinical trials as opposed to clinical studies"
--